# Impact of Brewing Methods on Total Phenolic Content (TPC) in Various Types of Coffee

**DOI:** 10.3390/molecules25225274

**Published:** 2020-11-12

**Authors:** Ewa Olechno, Anna Puścion-Jakubik, Renata Markiewicz-Żukowska, Katarzyna Socha

**Affiliations:** Department of Bromatology, Medical University of Bialystok, Mickiewicza 2D Street, 15-222 Bialystok, Poland; ewaolechno1996@gmail.com (E.O.); renmar@poczta.onet.pl (R.M.-Ż.); katarzyna.socha@umb.edu.pl (K.S.)

**Keywords:** Arabica, Robusta, coffee brewing, total phenolic content

## Abstract

Coffee is a widely consumed beverage, both in Europe, where its consumption is highest, and on other continents. It provides many compounds, including phenolic compounds. The aim of the study was to assess the effect of various brewing methods on the total phenolic content (TPC) in the infusion. Research material comprised commercially available coffees: Instant Arabica and Robusta, freshly ground Arabica and Robusta (immediately prior to the analysis), ground Arabica and Robusta, decaffeinated Arabica, and green Arabica and Robusta. The following preparation methods were used: Pouring hot water over coffee grounds or instant coffee, preparing coffee in a percolator and using a coffee machine. Additional variables which were employed were water temperature (90 or 100 °C) and its type (filtered or unfiltered). In order to determine the impact of examined factors, 225 infusion were prepared. Total phenolic content was determined by the spectrophotometric method using the Folin-Ciocalteu reagent and the obtained results were expressed in mg gallic acid (GAE) per 100 g of brewed coffee. The highest value was obtained for 100% Arabica ground coffee prepared in a coffee percolator using unfiltered water at a temperature of 100 °C: 657.3 ± 23 mg GAE/100 g of infusion. High values were also observed for infusions prepared in a coffee machine, where the highest TPC value was 363.8 ± 28 mg GAE/100 g for ground Arabica. In turn, the lowest TPC was obtained for Arabica green coffee in opaque packaging, brewed with filtered water at a temperature of 100 °C: 19.5 ± 1 mg GAE/100 g of infusion. No significant effect of temperature and water type on the TPC within one type of coffee was observed. Due to its high content of phenolic compounds, Arabica coffee brewed in a coffee percolator should be the most popular choice for coffee drinkers.

## 1. Introduction

Coffee is a widely consumed beverage, drunk primarily for its stimulating effect and aroma. Due to its worldwide popularity, it occupies an important place in international trade. The 2018 data show that the total production of coffee beans exceeded 10 million tons, an increase of 4.6% compared to the previous year. The leading producer of coffee is Brazil, where 3.8 million tons of beans were produced in 2018 [1]. Many sources report that coffee was brewed for the first time in Ethiopia, the former Abyssinia [2,3,4].

There are over 100 species of coffee shrubs, but it is Arabica (*Coffea arabica* L.) and Robusta (*Coffea canephora* Pierre ex A. Froehner) that are of greatest commercial importance—they account for 95% of the world’s coffee production [1].

The chemical composition of coffee beans varies depending on factors such as species, region where they are grown and the type of soil found there, cultivation method, climate conditions as well as the further processing of the beans, i.e., the cleaning and roasting process. More than 1500 constituents of coffee beans have been identified: 30–40% are carbohydrates, 13% are oils and 4% are proteins, including free amino acids. Furthermore, coffee beans also contain essential oils and phenolic compounds. Among the polyphenols present in them, the following acids are distinguished: Chlorogenic (CLA) and caffeic acid. Moreover, coffee also contains vitamins, minerals and aroma compounds [4,5].

It has been proven that regular consumption of coffee in moderate amounts (3–4 cups a day) has no negative health effects and may even contribute to reducing the risk of cancer, cardiovascular, liver and neurodegenerative diseases, including multiple sclerosis, and may aid carbohydrate metabolism [5,6,7,8,9]. In addition, coffee consumption has a marginal anti-inflammatory effect by affecting C-reactive protein, as demonstrated in a study of 112 men and 132 women [10]. The effect is mainly attributed to the presence of phenols in coffee.

Polyphenolic compounds are common in the plant world where, as secondary metabolites of plants, they are found in every part of the plant—the roots, leaves, seeds, flowers, fruit, bark and woody parts. They have a varied structure, different molecular weight and varying, chemical, physical and biological properties. They can be found in vegetables, fruit, nuts, coffee, tea and cocoa [11].

Polyphenols exert an antioxidant effect and are, therefore, crucial to the contemporary human diet. They show prophylactic activity, support the body to fight free radicals, the excess of which contributes to the development of the so-called lifestyle diseases such as cardiovascular diseases and cancer. The antioxidant effect consists, primarily, of intensifying the dismutation of free radicals to compounds of far lower reactivity, direct interaction with ROS (reactive oxygen species) and their sweeping, chelation of metals, in particular iron, with pro-oxidative properties as well as stimulation or inhibition of different types of enzyme activity. They have also been shown to have immunomodulatory effects. Factors such as stress, poor nutrition, smoking and excessive alcohol consumption increase the need for these compounds [12,13,14].

Polyphenols are divided into six main groups according to the structure of their basic carbon skeleton: Flavonoids, phenolic acids, proanthocyanides, stilbenes, lignans and lignins [10]. The most important phenolic compounds present in coffee are caffeoylquinic acids (CQA), dicaffeoylquinic acids and caffeoylquin lactone. Total content of caffeoylquinic acid in brews ranges from 45.79 to 1662.01 mg/L [15]. Among other substances, CLA shows particularly high antioxidant activity. It is formed by combining the phenol group of quinic acid with the carboxyl group of caffeic acid. Compared to the free form of caffeic acid, the esters of the acids are characterized by enhanced antioxidant properties. CLA is present in both beans and leaves of the coffee tree, with green coffee having a higher content. The amount of the acid decreases as coffee beans mature. It has been demonstrated that CLA has a positive effect on glucose and lipid metabolism, and thus, reduces the risk of diabetes, obesity, cardiovascular and liver diseases and cancer [16,17,18].

There are many types of coffee available on the market which can be brewed using a variety of methods: By pouring water over coffee grounds or granules, in a coffee percolator or in different types of coffee machines. There are few studies in the available literature assessing the effect of the brewing method (coffee type, water type and temperature) on the TPC [19,20,21] in infusions from various types of coffee. Therefore, the aim of the present study was to conduct such an evaluation.

## 2. Results

Table 1 presents descriptive statistics of the total phenolic content (mg gallic acid (GAE) per 100 g) for all types of coffee without taking into account the division according to the following variables: Brewing method, temperature and water. It was shown that infusions obtained from 100% Robusta coffee had the highest average (226.9 ± 200 mg GAE/100 g).

### 2.1. Total Phenolic Content in 100% Arabica Coffee

Figure 1 shows the effect of different brewing methods on the TPC in infusions of 100% Arabica coffee. The highest average TPC value was obtained for coffees prepared in a coffee percolator (numbers 16 and 17). Among them, the highest TPC value was obtained for ground Arabica prepared with unfiltered water at a temperature of 100 °C: 657.4 ± 23 mg GAE/100 g infusion (number 16). It was significantly higher than the TPC of all other coffees. As for soluble coffee, all infusions prepared using different parameters (filtered or unfiltered water at 90 °C or 100 °C) achieved similar TPC values—the highest value was obtained for coffee prepared with filtered water at a temperature of 100 °C: 151.4 ± 6 mg GAE/100 g of infusion (number 3). The lowest value among all brewing methods was obtained for freshly ground coffee prepared using filtered water at a temperature of 90 °C: 54.1 ± 4 mg GAE/100 g of infusion (number 8).

### 2.2. Total Phenolic Content in 100% Robusta Coffee

Figure 2 shows the effect of different brewing methods on the TPC in infusions of 100% Robusta coffee. The highest average TPC value was obtained for coffees prepared in a coffee percolator. Among them, the highest value was obtained for freshly ground Robusta prepared with unfiltered water: 642.7 ± 56 mg GAE/100 g of infusion (number 11). The value was significantly higher than the total content of phenolic compounds in all other infusions. Ground coffee prepared in a coffee machine displayed a value of 342.1 ± 13 mg GAE/100 g of infusion (number 16). As for instant coffee, similar TPC values were obtained for all infusions—the highest was recorded for coffee prepared with filtered water at 100 °C: 177.7 ± 2 mg GAE/100 g of infusion (number 3). Importantly, instant coffee outperformed ground coffee and freshly ground coffee prepared in the same manner. The lowest value for a ground coffee infusion was: 58.3 ± 2—unfiltered water, 90 °C—number 13.

### 2.3. Total Phenolic Content in Arabica Decaffeinated Coffee

Figure 3 shows the effect of different brewing methods on the TPC in infusions of decaffeinated Arabica coffee. The highest average value of TPC was obtained for coffees prepared in a coffee percolator. Among them, the highest value was obtained for freshly ground coffee prepared with filtered water at a temperature of 100 °C: 395.4 ± 36 mg GAE/100 g (number 10). It was a significantly higher value compared to the TPC of all other infusions. However, in the case of coffee made in a machine, a higher value was recorded for ground coffee: 266.4 287.4 ± 38 mg GAE/100 g of infusion (number 16). The TPC in all instant coffees was at a similar level—the highest value was found for coffee prepared with unfiltered water at a temperature of 100 °C (number 1): 153.1 ± 3 mg GAE/100 g.

### 2.4. Total Phenolic Content in Green Arabica Coffee

Figure 4 shows the effect of different brewing methods on the TPC in infusions of ground green Arabica. The highest average TPC was found for coffees prepared in a coffee percolator. Among them, the highest value was obtained for the infusion prepared with filtered water at a temperature of 100 °C: 257.0 ± 57 mg GAE/100 g of infusion (number 7). This was a higher value compared to the TPC of all other infusions. Coffee prepared in a coffee machine displayed values which were almost 40% lower: 161.0 ± 14 mg GAE/100 g of infusion (number 5). The lowest values among all brewing methods were recorded for coffee infusions prepared by pouring water from a kettle over coffee grounds at a temperature of 90 or 100 °C. Of these, the lowest TPC value was found for ground coffee prepared with filtered water at 100 °C: 19.5 ± 1 mg GAE/100 g infusion (number 3).

### 2.5. Total Phenolic Content in Green Arabica Coffee in Transparent Packaging

Figure 5 shows the effect of different brewing methods on the TPC in infusions of Arabica green ground coffee in transparent packaging. The highest average value of TPC was found for coffee prepared in a coffee percolator: 349.1 ± 29 mg GAE/100 g (number 6). This value was significantly higher compared to all other infusion preparation methods. For coffee made in a coffee machine, the value was 209.7 ± 12 mg GAE/100 g brew (number 7). The lowest values among all brewing methods were obtained for coffees prepared by pouring water from a kettle over coffee grounds at a temperature of 90 °C or 100 °C—among them, the lowest TPC was recorded for an infusion prepared with filtered water at a temperature of 90 °C 25.3 ± 7 mg GAE/100 g of infusion (number 4).

### 2.6. Total Phenolic Content in Green Robusta Coffee

Figure 6 shows the effect of different brewing methods on the TPC in infusion of Robusta green coffee. The highest TPC was found for coffee prepared in a coffee percolator: 253.3 ± 30 mg GAE/100 g (number 5) and in a coffee machine, using filtered water at a temperature of 100 °C: 279.0 ± 73 mg GAE/100 g (number 7). These values were significantly higher compared to the TPC of all other Robusta green coffee infusions prepared in this study. The lowest TPC values among all brewing methods were obtained for coffees prepared by pouring water from a kettle at a temperature of 90 °C or 100 °C—the lowest value was recorded for an infusion prepared with filtered water at a temperature of 90 °C: 37.1 ± 23 mg GAE/100 g of infusion (number 2).

### 2.7. Daily Consumption of Polyphenols with Coffee

Daily coffee consumption was assumed to be four cups, and the highest TPC value was selected for each type of coffee. It was demonstrated that with infusions prepared from Arabica coffee, 2628.8 mg of phenolic compounds (expressed as mg GAE) were provided, while with infusions prepared from Robusta coffee: 2553.6 mg GAE. The smallest amount of phenolic compounds was provided by green Arabica in opaque packaging (936.4 mg GAE) (Figure 7).

### 2.8. Principal Components Analysis (PCA)

The purpose of the PCA analysis is to transform the initial variables into new variables (principal components) to build a model describing the relationships between the studied features.

The inclusion of the Kaiser criterion leaves for interpretation components with eigenvalues above 1. Therefore, in the present study the model would have taken into account only two components (method and type), which would have explained only 63.9% of the variance. However, we assumed that the model should explain at least 75% of the variance, and therefore, the constructed model consisted of three components (method, type and water). The model explained 82.2% of the variance—the first component explained 42.21%, the second 21.65%, the third 18.34%. Figure 8 shows the projection of the factorial coordinates of the cases on the plane: Figure 8a—PC1 and PC2, Figure 8b—PC1 and PC3, while Figure 8c—PC2 and PC3. For the first component the variable ‘method’ (pouring water, percolator or coffee machine) was characterized by a high eigenvector, for the second component—‘type of coffee’ (instant, coffee beans or ground coffee), for the third component—‘water’ (filtered or unfiltered). PC2 and PC3 were shown to be responsible for discriminating primarily between Arabica and Robusta coffees from decaffeinated Arabica, green Arabica, green Robusta and green Arabica in transparent packaging. Figure 9 shows the projection of factor coordinates of the variables on the plane: Figure 9a—PC1 and PC2, Figure 9b—PC1 and PC3, Figure 9c—PC2 and PC3. The further the point is from the center, the greater the correlation of the variable with the factor axis and the better the representation on the chart by the factors. This indicates which variable is correlated with a given factor. The higher the absolute value of the factor loading, the higher the correlation of the variable with the component and the closer the proximity of the points to the circle on the plot—in the case of Figure 9a,b it is ‘method,’ and in the case of Figure 9c it is ‘type’ and ‘water’.

## 3. Discussion

An excess of free radicals in the body is one of the causes of lifestyle diseases such as cancer or diseases of the circulatory system. Therefore, it is essential that the human diet contains, among other nutrients, phenolic compounds [13]. Coffee is one of the dietary sources of these compounds [16,17].

Polyphenol content in plant is influenced by environmental factors such as plant variety and species, degree of maturity during harvest, climate conditions, including sunlight and rainfall, and soil conditions. Technological processes such as food storage and processing, including thermal processing, also play a crucial role [22].

An important factor that influences the prophylactic qualities of coffee is the content of phenolic compounds in the final coffee infusion intended for consumption. In addition to examining the impact of the preparation method, the present study also investigated the influence of temperature and type of water on the total phenolic compound content in the infusion. It was demonstrated that infusions prepared from the same type of coffee but using different brewing methods differed in terms of total phenolic compound content.

For coffees prepared in a percolator, the highest TPC in 100 g of infusion was obtained for the infusion of ground coffee: 657.4 ± 23 mg GAE/100 g of coffee infusion (ground Arabica, unfiltered water, 100 °C) and the lowest: 253.3 ± 30 mg GAE/100 g of coffee infusion (green Robusta coffee, unfiltered water, 100 °C).

High values were also obtained for infusions prepared in a coffee machine. Among the infusions obtained using this method, the highest value was found for ground Arabica: 363.8 ± 28 mg GAE/100 g of infusion, and the lowest for green Arabica in opaque packaging: 161.0 ± 14 mg GAE/100 g of infusion. The results reported by other authors differ from those obtained in the present paper [23,24].

There are few studies in the available literature that assess the effect of the coffee brewing method on the polyphenol content in the infusion. Merecz et al. (2018) studied Arabica, Robusta and green coffee in their publication. Coffee infusions were prepared by pouring hot water (90 °C) over ground beans, using a coffee percolator and a coffee machine. The authors of the study did not show any significant differences between the brewing method and coffee type, and the content of polyphenols. As in the present study, the highest polyphenol content was recorded for Arabica coffee infusion prepared in a coffee percolator. The authors, however, used a different method of presenting the results—in µg of caffeic acid per 100 µL of coffee brew. It is also worth noting that the brewing methods used by the authors differed from those utilized in the present study. In the current investigation, hot water was used and the coffee tank was filled to its maximum capacity (10 g). The way in which consumers brew coffee. On the other hand, in a study by Merecz et al. (2018), 2 g of coffee and cold water was used, which could have affected the degree of substance extraction in the infusion [20]. Therefore, it is not possible to perform an accurate comparison of the results due to differences in the methods used, but the conclusions drawn from both studies are similar.

Pressure, which increases the extraction of substances into the brew, may have an impact on the enhanced content of phenolic compounds in infusions prepared in a coffee percolator and a coffee machine [25].

In the case of instant coffee, it was possible to prepare the infusion only by pouring water over the product, which is the method used by consumers. The highest value among the tested instant coffees was found in the Robusta infusion prepared using filtered water at a temperature of 100 °C: 177.65 mg GAE/100 g of infusion, while the lowest value was noted for Arabica coffee made with unfiltered water at a temperature of 90 °C: 146.28 mg GAE/100 g of coffee infusion. The results obtained for instant coffees were significantly lower in comparison to those for coffee brewed in a coffee percolator and a coffee machine, but higher than those for ground and freshly ground coffee prepared in the same manner.

When comparing instant coffee to ground coffee, it is worth mentioning how the former is produced. Production consists of evaporating water from the prepared coffee extract. There are two main methods: Spray and freeze-drying. The first involves using a hot air stream on the final coffee brew, which produces the so-called agglomerated coffee in the form of fine granules or powder. In turn, the production of freeze-dried coffee consists of drying a frozen coffee extract [18]. Both methods involve the removal of excess water, and therefore, the concentration of ingredients that make up the brew increases. This may be a factor which contributes to the enhanced content of phenolic compounds in instant coffees, compared to ground and freshly ground coffees.

In a study by Chłopicka et al. (2015), ground, freshly ground and instant coffee infusions prepared by pouring boiling water and using an espresso machine were examined. The highest TPC was found in instant coffee (Tchibo Classic Family, Hamburg, Germany) prepared by pouring boiling water over the product: 836 mg GAE/L infusion (83.6 mg GAE per 100 g infusion). The conclusions from the present investigation and the study by Chłopicka et al. (2015) differ [18]. In this study, coffee brewed in an espresso machine was characterized by a higher TPC than instant coffee.

However, it is worth emphasizing that both studies demonstrated higher GAE equivalent values in instant coffees in comparison to ground and freshly ground coffee [18]. It is surprising that instant coffee contains more polyphenolic compounds than ground and freshly ground coffee prepared in the same way. According to literature reports, instant coffee also contains more oxalic acid in comparison to ground coffee—an average of 19 mg/1 g in a dry product according to Michalak-Majewska (2013) and 21–28 mg/1 g of dry product according to Sperkowska (2010). For individuals suffering from kidney stones, drinking several cups of coffee a day will have a significant effect on their mean daily oxalate intake [26,27].

The present study also investigated the value of TPC in the infusion obtained from decaffeinated Arabica. This type of coffee is the product of the decaffeination process, which involves adding a solvent to raw coffee beans and then removing it. Currently, in order to eliminate caffeine, chemical solvents such as methylene chloride and ethyl acetate, supercritical gases, mainly carbon dioxide, water and activated carbon are used [28,29]. Coffee can be called ‘decaffeinated’ when the content of anhydrous caffeine is less than or equal to 0.3% of the dry matter derived from coffee [30].

In the present study, decaffeinated Arabica was characterized by a lower TPC than regular Arabica. Both types of coffee demonstrated the highest content of polyphenolic compounds when prepared in a coffee percolator—for decaffeinated Arabica it was: 395.4 ± 36 mg GAE/100 g of coffee brew (freshly ground coffee, filtered water, 100 °C), while for Arabica it was: 657.4 ± 23 mg GAE/100 g brew (ground coffee, unfiltered water, 100 °C).

Other authors have not reported a significant impact of the decaffeination process on the content of polyphenols. As stated, the roasting process used has a more significant effect. In studies by Mills et al. (2013) and Hall et al. (2018), the content of polyphenolic compounds did not differ significantly between infusions obtained from decaffeinated coffee and those from other coffees [31,32].

As for green coffee, the present study examined infusions made by pouring hot water over coffee, brewing it in a coffee percolator and in a coffee machine. Consumers usually prepare the infusion as recommended by the manufacturer, i.e., by pouring water over ground beans. However, in order to be able to compare the influence of all factors (brewing method, temperature, water type), a coffee percolator and a coffee machine were used.

Green coffee is made from raw, unroasted coffee beans. Many studies have shown its beneficial effect on human health, including carbohydrate and lipid metabolism and blood pressure [33,34,35]. The most important ingredient in coffee beans is CLA. It has been demonstrated that the roasting process may affect its content in coffee beans [33,36,37].

In the present study, infusions prepared from green coffee did not display the highest TPC values. The highest TPC among green coffees infusions was found for Arabica in transparent packaging brewed in a coffee percolator using filtered water: 349.10 ± 29 mg GAE/100 g of coffee infusion, and the lowest for Arabica in an opaque bag, prepared with filtered water at 100 °C: 19.50 ± 1 mg GAE/100 g of coffee infusion. Initially, when planning the experiment, green Arabica was purchased in transparent packaging, but due to the potential influence of light on the content of phenolic compounds, green Arabica in opaque packaging from other manufacturers was also tested. It was established that the packaging did not have a significant impact on the final TPC. Surprisingly, coffee in transparent packaging had even higher TPC than coffee in opaque packaging. Both the origin conditions of production and storage of beans could play a role here. Additionally, it was observed that the green coffee available on the Polish market does not have a uniform grain structure, which may affect the extraction of polyphenolic substances into the brew.

A study by Wolska et al. (2017) also found that an infusion prepared from green coffee contained fewer polyphenols. The authors examined five methods of brewing, but only two of them corresponded to the methods used in the present study. In both studies, differences were noticed between the TPC in coffee brewed in a coffee machine and infusions prepared by pouring boiling water over coffee grounds. Wolska et al. (2017) used 1.5 g of coffee and 150 mL of water to prepare infusions. The time of infusing coffee with water was 5 min. Additionally, the authors used a filter coffee machine [24].

The same study also revealed that in comparison to the traditional Turkish coffee, green coffee was characterized by a higher TPC, which proves that brewing coffee in water at a temperature of 100 °C for a long time may affect the transfer of polyphenolic compounds to the brew [21]. It is worth noting that in publications describing the content of polyphenolic compounds per dry matter of the product, green coffee was demonstrated to have the highest antioxidant potential compared to roasted coffees [37,38,39].

The coffee roasting process takes place at temperatures exceeding 200 °C. Roasting affects the color of beans and the physical and chemical properties of coffee [38]. During the process, compounds that affect the taste and aroma of coffee are formed: Volatile and aromatic substances, but also products of the Maillard reaction, including acrylamide with potential carcinogenic, mutagenic and reproductive-reducing properties [40,41]. Moon et al. (2009) showed that the content of CLA in coffee beans decreases with the intensity of roasting—roasting at 230 °C for 12 min causes a decrease in CLA content by approximately 50% and roasting at 250 °C for 21 min leaves only traces of it in coffee beans [37]. Szymanowska et al. (2014) also observed that among all tested coffees with a different degree of roasting, green coffee was characterized by the highest TPC: expressed as 3.28 and 3.40 g of chlorogenic acid/100 g of dry matter [38]. A study by Mills et al. (2013) revealed that medium roast coffee has higher antioxidant potential than green coffee, but lower than heavily roasted coffee [31].

However, some studies have demonstrated that the antioxidant potential of coffee beans decreases only marginally during the roasting process. This may be due to the fact that, despite the partial decomposition of polyphenolic compounds during the process of roasting beans, new substances exhibiting antioxidant activity are formed, which compensates for the loss of polyphenols. These are mainly melanoidins formed during the Maillard reaction [38]. Their antioxidant properties have been demonstrated in living organisms and utilized in food manufacturing by enabling extension of the shelf life of food [42,43]. Moreover, their antibacterial activity has been observed, particularly, towards *Bacillus cereus*, *Escherichia coli*, *Helicobacter pylori*, *Proteus mirabilis*, *Pseudomonas aeruginosa*, *Salmonella typhimurium* and *Staphylococcus aureus*. The compounds may also stimulate the growth of intestinal microbiota and lower cholesterol levels [42,43,44,45,46].

When analyzing the influence of temperature (90 °C and 100 °C) and water type (filtered and unfiltered), no statistically significant differences were found in the total content of phenolic compounds. Chłopicka et al. (2015) demonstrated that the use of highly mineralized water reduces polyphenol content in the infusion [18]. This may be due to the formation of complexes between polyphenolic compounds and minerals.

The present study demonstrated that in order to obtain the highest content of phenolic compounds in the infusion, the recommended method of brewing coffee is in a coffee percolator and a coffee machine. No statistically significant differences between the type and temperature of water used were found.

In 2015, data on the estimated intake of phenolic compounds by the population of Brazil were published. It averaged 460.15 ± 341.33 and almost half of this amount was consumed with beverages (225.36 ± 191.1). Coffee consumption covered 83.58% of this amount [47]. Research conducted in Poland in 2015 revealed that the average consumption of phenolic compounds among men was 2357 ± 477, and among women 1862 ± 152. Of this amount, beverages provided 33% and 37% of the daily consumption of these compounds for men and women, respectively [48].

## 4. Materials and Methods

### 4.1. Coffee Materials

The research material consisted of samples of coffee purchased from supermarkets and health food stores (Białystok, Poland), and from online shops.

The following types of coffee (100% Arabica or 100% Robusta) were tested (n = 36): Instant Arabica (n = 3), Arabica beans (n = 3), ground Arabica (n = 3), instant Robusta (n = 3), Robusta beans (n = 3), ground Robusta (n = 3), decaffeinated instant Arabica (n = 3), decaffeinated Arabica beans (n = 3), decaffeinated ground Arabica (n = 3), ground green Arabica (n = 3), ground green Arabica in transparent packaging (n = 3), ground green Robusta (n = 3).

Infusions were prepared using a combination of factors (coffee type, brewing method, water type, water temperature)—three infusions were made for each combination of factors, two samples were taken from each infusion for spectrophotometric determination.

### 4.2. Coffee Brewing

The following brewing methods were used to prepared infusions: pouring water boiled in a kettle over coffee grounds, a coffee percolator and a coffee machine. The infusions were prepared using filtered and unfiltered tap water. The influence of water temperature (90 and 100 °C) was also analyzed. A total of 225 types of infusions were prepared using the above variables. These factors were taken into consideration because of different preferences of consumers regarding coffee brewing methods.

In order to make simple coffee infusions, 2 g of coffee grounds (which corresponds to one teaspoon according to the literature data) were weighed and 150 mL of water was poured over them (average capacity of one cup). The infusion time was 5 min [17].

To prepare the infusion in a coffee percolator (Bialetti, Coccaglio, Italy), 10 g of coffee was weighed (the maximum amount that could fit into the strainer) and 100 mL of hot water was poured into the lower container (maximum capacity of the container). The preparation time of the infusion was 3 min.

The preparation of coffee in a coffee machine involved placing 7 g of ground coffee in the container of the coffee machine (Saeco Royal Professional, Gaggio Montano, Italy) or, in the case of whole beans, preparing an infusion from 7 g of the product. The obtained espresso had a volume of 70 mL.

### 4.3. Total Phenolic Content Analysis

In order to prepare samples, 0.5 mL of infusion was collected in the case of coffee machine and coffee percolator, and 1 mL for the remaining coffees (volumes were determined experimentally on the basis of the standard curve range). Then, the mixture was made up to 10 mL with distilled water and centrifuged for 5 min, speed: 2000 rpm (MPW-340 centrifuge, Mechanika Precyzyjna, Poland). All determinations were performed in duplicate.

Following sample centrifugation, 0.25 mL of supernatant was withdrawn, 1.25 mL of 0.2 N Folin-Ciocalteu reagent was added, followed by thorough mixing for 5 min. Then, 1 mL of Na_2_CO_3_ solution was added, mixed and incubated for 2 h at room temperature. The absorbance was measured against water as a blank—wavelength 760 nm (U-2001, Hitachi Instruments Inc., Tokyo, Japan). The concentration as mg of GAE/100 g was read from the standard curve [mg GAE/100 g of infusion] [19,20,21].

### 4.4. Estimation of Daily Intake of Phenolic Compounds

In order to estimate the daily consumption of phenolic compounds with coffee, the infusion with the highest TPC value was selected for each type of coffee. The value was multiplied by four cups (average coffee consumption per day) [7].

### 4.5. Statistical Analysis

The results of analyses of the total content of phenolic compounds depending on the method of brewing in various types of coffee were statistically analyzed using Statistica 13.3 and Excel.

The normality of the data distribution was verified by appropriate tests: Shapiro-Wilk, Kołmogorow-Smirnow and Lilliefors. The Wilcoxon sequence test was used to assess differences in TPC values between individual coffee brews. The level of significance in the analyses was *p* < 0.05. Additionally, descriptive statistics were determined: arithmetic means, standard deviations, minimum and maximum values, medians as well as lower (Q1) and upper (Q3) quartiles and the range between them (IQR). Additionally, Principal Components Analysis (PCA) was performed to reduce data and identify main factors responsible for variability.

## 5. Conclusions

Brewing methods—using a coffee machine, a coffee percolator and pouring water from a kettle—have a significant impact on the total phenolic content in a coffee brew, which was confirmed by the PCA analysis. The most recommended method for all ground and freshly ground coffee is a coffee percolator. TPC values for instant coffee infusions were higher in comparison to those for ground and freshly ground coffee brewed by pouring water over coffee grounds. The infusion prepared from green coffee was characterized by a lower TPC value than that from roasted coffee, while higher TPC values were observed for coffee in transparent packaging in comparison to opaque packaging. Water temperature and type of water used had no effect on TPC.

## Figures and Tables

**Figure 1 molecules-25-05274-f001:**
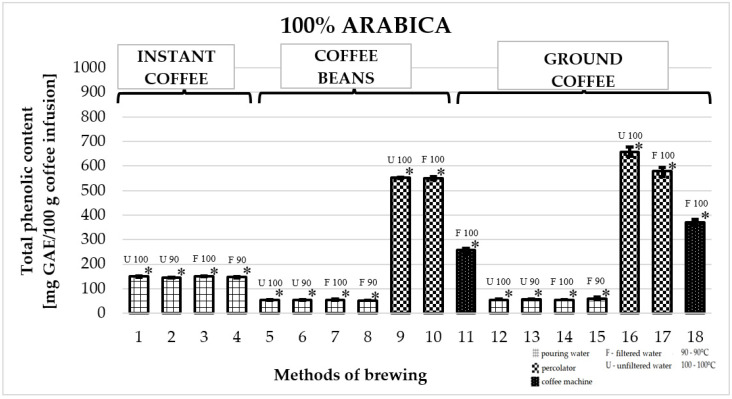
Median of total phenolic content as gallic acid equivalent [mg GAE/100 g of infusion] in 100% Arabica coffee. * *p*_1/5,6,7,8,9,10,11,12,13,14,15,16,17,18_; * *p*_2/5,6,7,8,9,10,11,12,13,14,15,16,17,18_; * *p*_3/5,6,7,8,9,10,11,12,13,14,15,16,17,18_; * *p*_4/5,6,7,8,9,10,11,12,13,14,15,16,17,18_; * *p*_5/1,2,3,4,9,10,11,15,16,17,18_; * *p*_6/1,2,3,4,9,10,11,15,16,17,18_; * *p*_7/1,2,3,4,9,10,11,15,16,17,18_; * *p*_8/1,2,3,4,9,10,11,16,17,18_; * *p*_9/1,2,3,4,5,6,7,8,11,12,13,14,15,16,18_; * *p*_10/1,2,3,4,5,6,7,8,11,12,13,14,15,16,18_; * *p*_11/1,2,3,4,5,6,7,8,9,10,12,13,14,15,16,17,18_; * *p*_12/1,2,3,4,9,10,11,16,17,18_; * *p*_13/1,2,3,4,9,10,11,15,16,17,18_; * *p*_14/1,2,3,4,9,10,11,15,16,17,18_; * *p*_15/1,2,3,4,5,6,7,9,10,11,13,14,16,17,18_; * *p*_16/1,2,3,4,5,6,7,8,9,10,11,12,13,14,15,17,18_; * *p*_17/1,2,3,4,5,6,7,8,11,12,13,14,15,16,18_; * *p*_18/1,2,3,4,5,6,7,8,9,10,11,12,13,14,15,16,17_—statistically significant differences (* *p* < 0.05). Preparation of the infusion: **1:** pouring water, 100 °C, unfiltered water; **2:** pouring water, 90 °C, unfiltered water; **3:** pouring water, 100 °C, filtered water; **4:** pouring water, 90 °C, filtered water; **5:** pouring water, 100 °C, unfiltered water; **6:** pouring water, 90 °C, unfiltered water; **7:** pouring water, 100 °C, filtered water; **8:** pouring water, 90 °C, filtered water; **9:** coffee percolator, 100 °C, unfiltered water; **10:** coffee percolator, 100 °C, filtered water; **11:** coffee machine, 100 °C, filtered water; **12:** pouring water, 100 °C, unfiltered water; **13:** pouring water, 90 °C, unfiltered water; **14:** pouring water, 100 °C, filtered water; **15:** pouring water, 90 °C, filtered water; **16:** coffee percolator, 100 °C, unfiltered water; **17:** coffee percolator, 100 °C, filtered water; **18:** coffee machine, 100 °C, filtered water.

**Figure 2 molecules-25-05274-f002:**
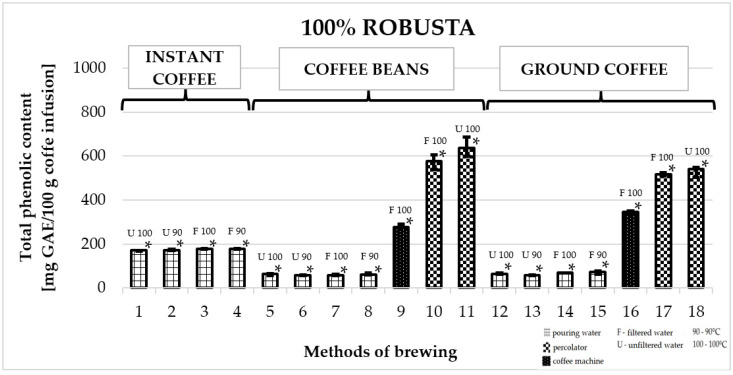
Median of total phenolic content as gallic acid equivalent [mg GAE/100 g of infusion] in 100% Robusta coffee. * *p*_1/3,4,5,6,7,8,9,10,11,12,13,14,15,16,17,18_; * *p*_2/3,4,5,6,7,8,9,10,11,12,13,14,15,16,17,18_; * *p*_3/1,2,5,6,7,8,9,10,11,12,13,14,15,16,17,18_; * *p*_4/1,2,5,6,7,8,9,10,11,12,13,14,15,16,17,18_; * *p*_5/1,2,3,4,9,10,11,16,17,18_; * *p*_6/1,2,3,4,9,10,11,12,14,15,16,17,18_; * *p*_7/1,2,3,4,9,10,11,12,14,16,17,18_; * *p*_8/1,2,3,4,9,10,11,16,17,18_; * *p*_9/1,2,3,4,5,6,7,8,10,11,12,13,14,15,16,17,18_; * *p*_10/1,2,3,4,5,6,7,8,9,11,12,13,14,15,16_; * *p*_11/1,2,3,4,5,6,7,8,9,10,12,13,14,15,16,17,18_; * *p*_12/1,2,3,4,6,7,9,10,11,13,14,16,17,18_; * *p*_13/1,2,3,4,9,10,11,12,14,15,16,17,18_; * *p*_14/1,2,3,4,6,7,9,10,11,12,13,16,17,18_; * *p*_15/1,2,3,4,6,9,10,11,13,16,17,18_; * *p*_16/1,2,3,4,5,6,7,8,9,10,11,12,13,14,15,17,18_; * *p*_17/1,2,3,4,5,6,7,8,9,11,12,13,14,15,16,18_; * *p*_18/1,2,3,4,5,6,7,8,9,11,12,13,14,15,16_—statistically significant differences (* *p* < 0.05). Preparation of the infusion: **1:** pouring water, 100° C, unfiltered water; **2:** pouring water, 90° C, unfiltered water; **3:** pouring water, 100° C, filtered water; **4:** pouring water, 90° C, filtered water; **5:** pouring water, 100 °C, unfiltered water; **6:** pouring water, 90 °C, unfiltered water; **7:** pouring water, 100 °C, filtered water; **8:** pouring water, 90 °C, filtered water; **9:** coffee machine, 100 °C, filtered water; **10:** coffee percolator, 100 °C, filtered water; **11:** coffee percolator, 100 °C, unfiltered water; **12:** pouring water, 100 °C, unfiltered water; **13:** pouring water, 90 °C, unfiltered water; **14:** pouring water, 100 °C, filtered water; **15:** pouring water, 90 °C, filtered water; **16:** coffee machine, 100 °C, filtered water; **17:** coffee percolator, 100 °C, filtered water; **18:** coffee percolator, 100 °C, unfiltered water.

**Figure 3 molecules-25-05274-f003:**
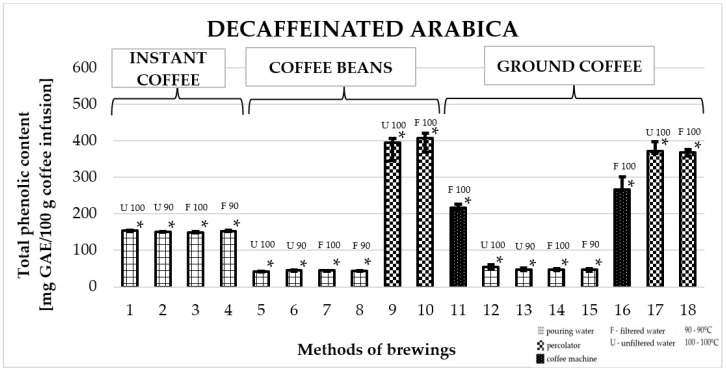
Median of total phenolic content as gallic acid equivalent [mg GAE/100 g of infusion] in 100% decaffeinated Arabica coffee. * *p*_1/3,5,6,7,8,9,10,11,12,13,14,15,16,17,18_; * *p*_2/5,6,7,8,9,10,11,12,13,14,15,16,17,18_; * *p*_3/1,5,6,7,8,9,10,11,12,13,14,15,16,17,18_; * *p*_4/5,6,7,8,9,10,11,12,13,14,15,16,17,18_; * *p*_5/1,2,3,4,6,7,8,9,10,11,12,13,15,16,17,18_; * *p*_6/1,2,3,4,5,9,10,11,12,13,15,16,17,18_; * *p*_7/1,2,3,4,5,9,10,11,12,15,16,17,18_; * *p*_8/1,2,3,4,5,9,10,11,12,15,16,17,18_; * *p*_9/1,2,3,4,5,6,7,8,10,11,12,13,14,15_; * *p*_10/1,2,3,4,5,6,7,8,9,11,12,13,14,15,16,18_; * *p*_11/1,2,3,4,5,6,7,8,9,10,12,13,15,16,17,18_; * *p*_12/1,2,3,4,5,6,7,8,9,10,11,15,16,17,18_; * *p*_13/1,2,3,4,5,6,9,10,11,15,16,17,18_; * *p*_14/1,2,3,4,9,10,15,16,17,18_; * *p*_15/1,2,3,4,5,6,7,8,9,10,11,12,13,14,16,17,18_; * *p*_16/1,2,3,4,5,6,7,8,10,11,12,13,14,15,17,18_; * *p*_17/1,2,3,4,5,6,7,8,11,12,13,14,15,16,18_; * *p*_18/1,2,3,4,5,6,7,8,10,11,12,13,14,15,16,17_—statistically significant differences (* *p* < 0.05). Preparation of the infusion: **1:** pouring water, 100 °C, unfiltered water; **2:** pouring water, 90 °C, unfiltered water; **3:** pouring water, 100 °C, filtered water; **4:** pouring water, 90 °C, filtered water; **5:** pouring water, 100 °C, unfiltered water; **6:** pouring water, 90 °C, unfiltered water; **7:** pouring water, 100 °C, filtered water; **8:** pouring water, 90 °C, filtered water; **9:** coffee percolator, 100 °C, unfiltered water; **10:** coffee percolator, 100 °C, filtered water; **11:** coffee machine, 100 °C, filtered water; **12:** pouring water, 100 °C, unfiltered water; **13:** pouring water, 90 °C, unfiltered water; **14:** pouring water, 100 °C, filtered water; **15:** pouring water, 90 °C, filtered water; **16:** coffee machine, 100 °C, filtered water; **17:** coffee percolator, 100 °C, unfiltered water; **18:** coffee percolator, 100 °C, filtered water.

**Figure 4 molecules-25-05274-f004:**
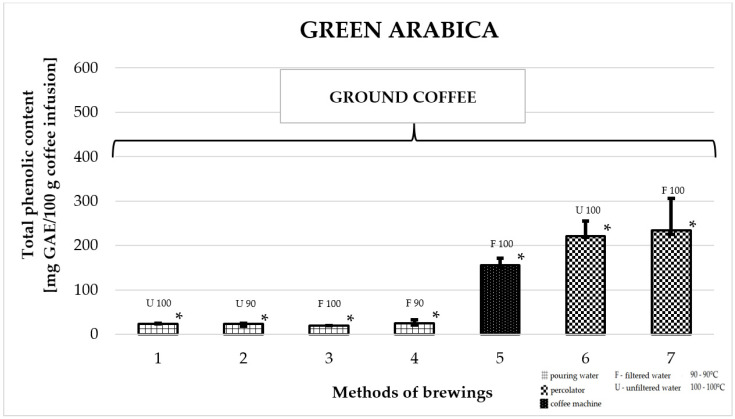
Median of total phenolic content as gallic acid equivalent [mg GAE/100 g of infusion] in 100% green Arabica coffee. * *p*_1/3,5,6,7_; * *p*_2/5,6,7_; * *p*_3/1,5,6,7_; * *p*_4/5,6,7_; * *p*_5/1,2,3,4,6,7_; * *p*_6/1,2,3,4,5_; * *p*_7/1,2,3,4,5_—statistically significant differences (* *p* < 0.05). Preparation of the infusion: **1:** pouring water, 100 °C, unfiltered water; **2:** pouring water, 90 °C, unfiltered water; **3:** pouring water, 100 °C, filtered water; **4:** pouring water, 90 °C, filtered water; **5:** coffee machine, 100 °C, filtered water; **6:** coffee percolator, 100 °C, unfiltered water; **7:** coffee percolator, 100 °C, filtered water.

**Figure 5 molecules-25-05274-f005:**
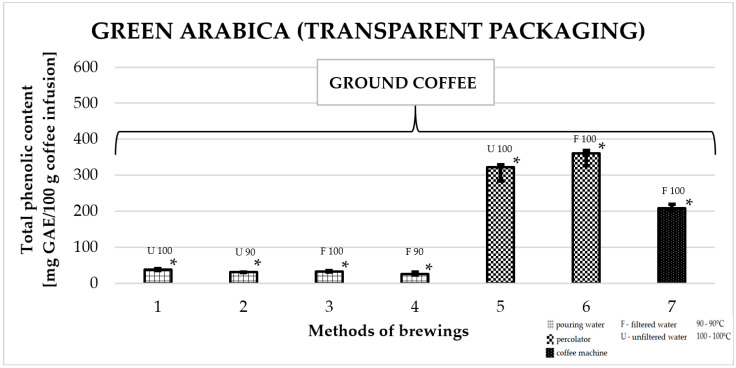
Total phenolic content as gallic acid equivalent [mg GAE/100 g of infusion] in 100% green Arabica coffee in transparent packaging. * *p*_1/2,4,5,6,7_; * *p*_2/1,5,6,7_; * *p*_3/5,6,7_; * *p*_4/1,5,6,7_; * *p*_5/1,2,3,4,7_; * *p*_6/1,2,3,4,7_; * *p*_7/1,2,3,4,5,6_—statistically significant differences (* *p* < 0.05). Preparation of the infusion: **1:** pouring water, 100 °C, unfiltered water; **2:** pouring water, 90 °C, unfiltered water; **3:** pouring water, 100 °C, filtered water; **4:** pouring water, 90 °C, filtered water; **5:** coffee percolator, 100 °C, unfiltered water; **6:** coffee percolator, 100 °C, filtered water; **7:** coffee machine, 100 °C, filtered water.

**Figure 6 molecules-25-05274-f006:**
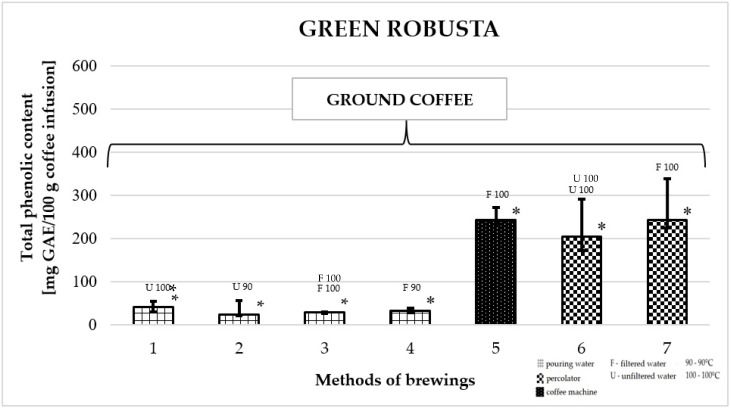
Total phenolic content as gallic acid equivalent [mg GAE/100 g of infusion] in 100% green Robusta coffee. * *p*_1/5,6,7_; * *p*_2/5,6,7_; * *p*_3/5,6,7_; * *p*_4/5,6,7_; * *p*_5/1,2,3,4_; * *p*_6/1,2,3,4_; * *p*_7/1,2,3,4_—statistically significant differences (* *p* < 0.05). Preparation of the infusion: **1:** pouring water, 100 °C, unfiltered water; **2:** pouring water, 90 °C, unfiltered water; **3:** pouring water, 100 °C, filtered water; **4:** pouring water, 90 °C, filtered water; **5:** coffee machine, 100 °C, filtered water; **6:** coffee percolator, 100 °C, unfiltered water; **7:** coffee percolator, 100 °C, filtered water.

**Figure 7 molecules-25-05274-f007:**
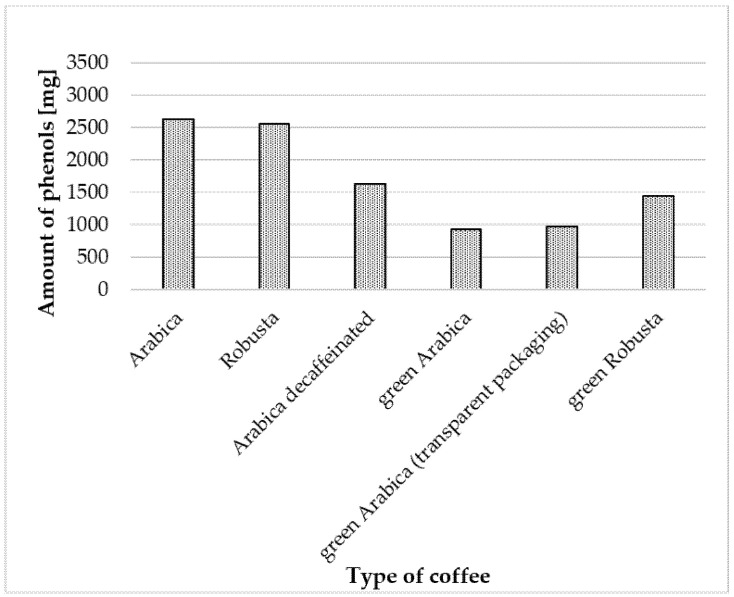
Daily consumption of phenolic compounds with four cups of coffee infusions, which had the highest TPC value.

**Figure 8 molecules-25-05274-f008:**
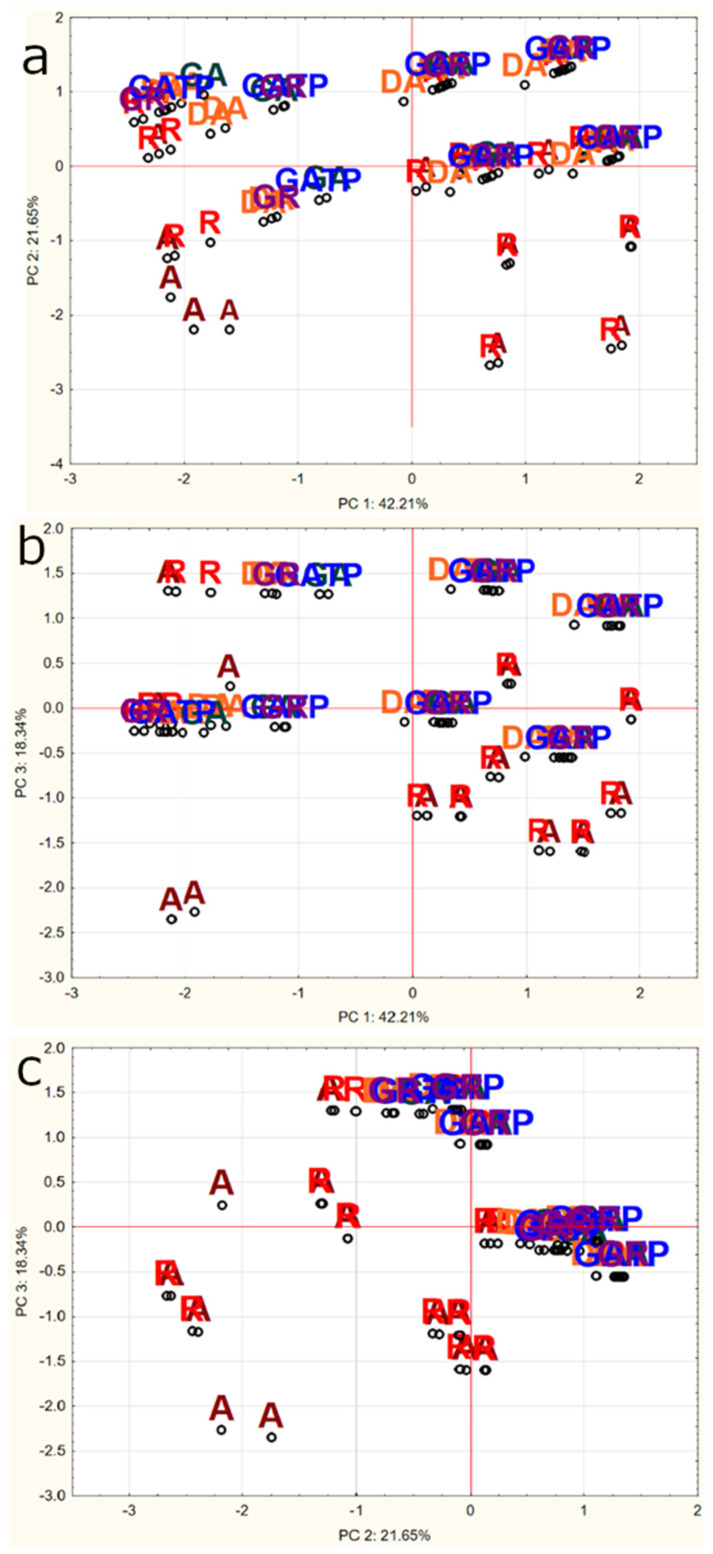
Plot of the principal component for the cases: (**a**) Principal Component 1 and 2, (**b**) Principal Component 1 and 3, (**c**) Principal Components 2 and 3. A—Arabica, DA—decaffeinated Arabica, GA—green Arabica, GATP—green Arabica in transparent packaging, GR—green Robusta, R—Robusta.

**Figure 9 molecules-25-05274-f009:**
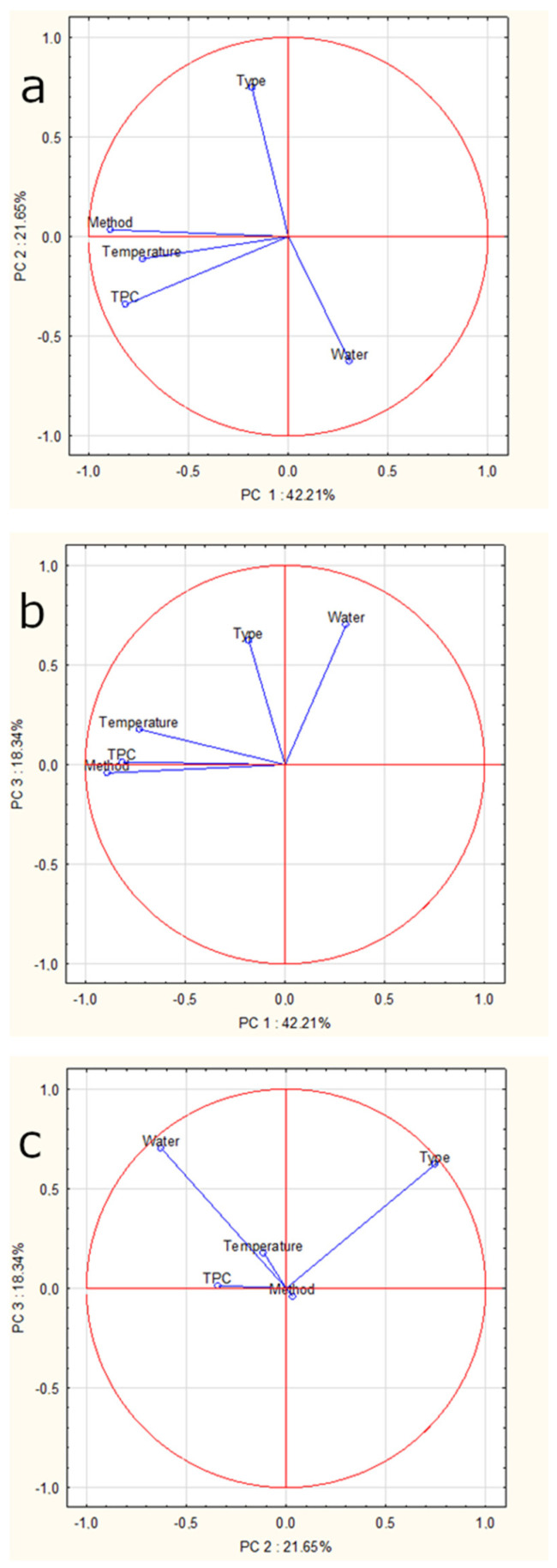
Plot of the principal component for the variables: (**a**) Principal Component 1 and 2, (**b**) Principal Component 1 and 3, (**c**) Principal Components 2 and 3.

**Table 1 molecules-25-05274-t001:** Descriptive statistics of total phenolic content (mg GAE/100 g) in all tested infusions of various types of coffee.

Type of Coffee	n	Average ± SD	Med.	Min.–Max.	Quartile Lower–Upper	IQR
100% Arabica	54	221.8 ± 212	145.8	50.4–683.2	55.5–371.3	315.8
100% Robusta	54	226.9 ± 200	171.6	53.4–711.2	62.8–347.1	284.3
decaffeinated Arabica	54	166.9 ± 134	150.0	38.8–430.4	45.6–266.4	220.8
green Arabica	21	106.5 ± 104	27.4	16.3–329.6	21.8–215.8	194.0
green Arabica (transparent packaging)	21	142.2 ± 135	39.7	15.6–378.2	31.2–271.0	239.8
green Robusta	21	129.0 ± 117	59.8	20.7–375.8	29.1–230.4	201.3

IQR—interquartile range.

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
