# Peer review of "Impact of Brewing Methods on Total Phenolic Content (TPC) in Various Types of Coffee"

_molecules, 2020, doi:10.3390/molecules25225274_

Round 1

Reviewer 1 Report

In this paper the significant impact of brewing methods of various types of coffee on the content of phenolic compounds has been studied.

The materials and methods are well described. Conclusions underline the importance of conducted investigations and novelty. 

The authors compared obtained results with other reports. Manuscript is prepared according to authors guidelines however a small recommendation is to improve the reference list i.e. to convert a full name of a proper cited journal into required abbraviated name (see Ref. 16, 17, 23).

Reviewer 2 Report

The manuscript by Olechno investigated and compared the effect of different brewing methods on the extraction of phenolic compounds in different types of coffee.

In my opinion, this study is lack of novelty and is basically based on spectrophotometric assay without a tentative identification of the phenolic compounds in the samples. There are plenty of papers investigating, even more thoroughly, the effect of brewing method on the bioactive compounds extraction in coffee.

General comments:

The results section should be greatly improved in the data presentation. For example is not necessary to report the Q1 and Q3 data (reduces the fluidity of reading). I suggest reporting data as average value with standard deviation in the text. The figures are difficult to read and the bar symbol are not easy to follow. Please use suprascript letter for the statistic, it is more easy to understand and gave an immediate picture of the statistical differences.

The inclusion of a principal component analysis may help to distinguish the effect of the variable on the extraction of phenolic compounds.

The discussion section is too long and not very informative.

Other comments:

Please focus introduction on the scientific aspect. Lines 37-46 could be deleted.

Lines 55-56: Quinic acid and alkaloids are not phenolic compounds. Please re-write the sentence.

Lines 78-80: The phenolic compounds in coffee have been extensively studied. The most important are caaffeoylquinic acids, dicaffeoylquinic acids and caffeoylquinic lactone. Please update the list deleting quinic acid and coffee acid.

Lines 89-91: There are numerous studies about the effect of brewing method on the extraction of phenolic compounds. Reference 18 is not appropriate.

Line 103: Median or average value?

Lines 106-107: “As for instant coffee, similar to each other TPC values were obtained for all samples”. Sentence not clear.

Lines 313-329. These sentences are out of topic. I suggest to delete them.

Lines 444-451: The description of the preparation of the calibration curve is not necessary. Please remove it from the paper.

Reviewer 3 Report

The article entitled " Impact of brewing methods on Total Phenolic 3 Content (TPC) in various types of coffee” describes the effect of various brewing methods on the total phenolic content (TPC) in the infusión of coffee.

The paper is well organized and the studies of how the thermal treatments can affect to the desirable and/or non desirable substances presents in food are essential and necessary from a scientific point of view.

But some aspects need to be modified.

In my opinion, the article can be published after Major Revision.

Comments

Pag. 10, line 245. Daily coffee consumption was assumed to be four cup. Why?

Pag.14, lines 418-425. In the Materials and Methods, the authors indicated “2.1. Coffee materials. The research material consisted of samples of coffee purchased in supermarkets and health food stores in Bialystok (Poland), and in online shops. The following types of coffee (100% Arabica or 100% Robusta) were tested (n = 36): instant Arabica (n = 3), Arabica beans (n = 3), ground Arabica (n = 3), instant Robusta (n = 3), Robusta beans (n = 3), ground Robusta (n = 3), decaffeinated instant Arabica (n = 3), decaffeinated Arabica beans (n  = 3), decaffeinated ground Arabica (n = 3), ground green Arabica (n = 3), ground green Arabica in 425 transparent packaging (n = 3), ground green Robusta (n = 3)”.

In my opinion, the number of samples is not sufficient for can be establish statistical differences into the different types of coffee. For this purpose, more samples are necessary.

Round 2

Reviewer 2 Report

The authors improved their manuscript and responded to all the criticisms raised by the reviewers 

Author Response

Dear Reviewer,

Thank you for all the advice that contributed to the improvement of the quality of our manuscript.
We are very thankfull.

Yours faithfully,
Authors

This manuscript is a resubmission of an earlier submission. The following is a list of the peer review reports and author responses from that submission.